# Access to Care for US Children with Co-Occurrence of Autism Spectrum Disorder and Epilepsy

**DOI:** 10.3390/children9071094

**Published:** 2022-07-21

**Authors:** Wanqing Zhang, Kelsey L. Thompson

**Affiliations:** 1Department of Health Sciences, School of Medicine, University of North Carolina at Chapel Hill, Chapel Hill, NC 27599, USA; 2Division of Speech and Hearing Sciences, School of Medicine, University of North Carolina at Chapel Hill, Chapel Hill, NC 27599, USA; kelsey_thompson@med.unc.edu

**Keywords:** autism spectrum disorder (ASD), epilepsy, access to care, co-occurring ASD and epilepsy (ASD-EP), inequity in ASD-EP

## Abstract

Epilepsy is a common comorbidity among children with autism spectrum disorder (ASD). There is a lack of understanding of the inequality in access to care for children with co-occurring ASD and epilepsy (ASD-EP). The purpose of this study is to examine key indicators for access to care and care coordination for children with ASD-EP in the US National Survey of Children’s Health (NSCH). Data were collected from the 2017–2019 NSCH. Our analytic sample included children with ASD without epilepsy (N = 2150), children with both ASD and epilepsy (N = 143), and children with epilepsy without ASD (N = 711). The dependent variables included important access to care indicators such as having usual sources of care, having adequate coverage, being frustrated in efforts to get service, and receiving care coordination. The independent variables included ASD-EP status, child demographics, and an intellectual disability (ID) diagnosis. Our results show that demographic characteristics such as sex, race, income level, and insurance type affect access to care. Inadequate access to healthcare was significantly higher among female children, children from low-income families, and children with ID. The access barriers among children with ASD-EP were more likely due to the interplay of multiple clinical and individual factors.

## 1. Introduction

Autism spectrum disorder (ASD) is a complex neurodevelopmental disorder with an etiology based in a combination of genetic and environmental influences. ASD is characterized by communication impairment, social interaction deficits, and stereotyped repetitive behaviors [1,2]. Children with ASD have a substantial burden of medical illness and tend to have multiple comorbidities requiring complex medical care [3,4,5]. Epilepsy is a common comorbidity in children with ASD that is characterized by recurrent unprovoked seizures [6,7,8]. A review of epidemiological studies indicated that epilepsy prevalence rates in children and young adults with ASD ranged from 2.4% to 26% [9]. While approximately 1% of children have epilepsy in the general population, 12.5% prevalence of epilepsy was observed in a population-based sample among children under age 17 [10]. A coordinated and comprehensive management approach is needed to consider the neurologic and socio-psychological consequences of ASD and epilepsy [6,11,12].

Previous studies have found increased utilization of costly health services among children with ASD [13,14]. Epilepsy is a concerning medical condition that could add to this burden and affect the lives of people with ASD. However, effectively managing and treating epilepsy can improve the quality of life of the children and their family greatly. Epilepsy care involves primary care physicians, the emergency department (ED), referral to neurologists/epileptologists, and care coordination among health professionals [15]. Children with both ASD and epilepsy have more involved medical care needs; thus, they require coordinated care from providers across disciplines to address more complicated symptoms. For example, ASD increases the risk of receiving epilepsy-related emergency care and hospitalizations [16]. There is currently a lack of understanding of inequality in access to care for children with co-occurring ASD and epilepsy. Therefore, there is an opportunity to improve access to care for underserved children and adolescents, to improve co-management of patients between primary care and specialty providers, and to improve care for those with co-occurrence of ASD and epilepsy.

A better understanding of disparities in access to care with respect to the dual diagnosis of ASD and epilepsy can provide insights into more effective management of epilepsy and prevention of seizure complications for children with co-occurrence of ASD and epilepsy. Our objective in this study was to examine whether there are disparities in the access to and utilization of health services between (a) children with co-occurrence of ASD and epilepsy, (b) children diagnosed with epilepsy only, and (c) children with ASD without epilepsy. We examined key indicators for access to care, care coordination, and parental frustration for children with ASD and/or epilepsy in the recent US National Survey of Children’s Health (NSCH). This present study used pooled data from the public use file of NSCH 2017–2019 to obtain a large database and conduct subgroup comparisons. This secondary data analysis aimed to describe factors associated with access to care in children with ASD and epilepsy, and based on the results, to provide public policy recommendations that could improve access to and coordinated care for vulnerable groups of children.

## 2. Materials and Methods

### 2.1. Data Source

The NSCH is a national survey of children under age 18 in the US. The survey is funded by the Maternal and Child Health Bureau with the US Census Bureau and is administered to parents or primary caregivers in all 50 US states and the District of Columbia with the goal of producing national estimates on the prevalence of health indicators as well as children’s experiences within the health care system [17]. A weighted probability sample was taken from the Census Bureau’s Master Address File, and one child in each household was randomly selected to be the subject of the survey. The NSCH oversamples children with special health care needs (e.g., ASD, developmental delays), and the survey respondent is the primary caregiver who is most familiar with the child’s health and health care. Since 2016, NSCH datasets have included questions about child demographics, access to and utilization of health care services, and special developmental services for children with ASD. We pooled data that was collected from June 2017 to January 2020 via a two-phase, self-administrated data collection design [18]. The NSCH dataset is available to the public and therefore does not contain identifiable information.

### 2.2. Participants and Measures

Children aged 2–17 years were included in this analysis. Our analytic sample was initially divided into two cohorts, children with ASD (N = 2293) and children with epilepsy (N = 854). The cohorts were further subdivided into children with ASD without epilepsy (N = 2150), children with both ASD and epilepsy (N = 143), and children with epilepsy without ASD (N = 711). Epilepsy diagnosis was determined with the question: “Has a doctor or other health care provider EVER told you that this child has Epilepsy or seizure disorder?” ASD status was determined based on the question that asked: “Has a doctor or other health care provider ever told you that this child has Autism or Autism Spectrum Disorder (ASD)?” The dual ASD-epilepsy group included all children where a “yes” response was given to both questions.

All analyzed outcomes in this study were based on parent report. Ten dichotomous dependent variables comprised: having usual sources of care (USC), having primary care physicians (PCP), needing referral, visiting the ED, having adequate coverage, having forgone health care, being frustrated in efforts to get service, receiving special services, receiving help on care coordination, and needing extra help for coordinating. To code the variables a 1 was given if the parent responded “yes” to the specific question and 0 if answered “no”. Having adequate coverage and parents’ frustration in efforts to get service were based on the following questions: “How often does this child’s health insurance offer benefits or cover services that meet this child’s needs?” and “During the past 12 months, how often were you frustrated in your efforts to get services for this child?” While both questions had 4 options (never, sometimes, usually, always), they were dichotomized into yes (usually or always) or no (never or sometimes). The question on forgone health care asked: “During the past 12 months, was there any time when this child needed health care but it was not received?” Parents were also asked “Did anyone help you arrange or coordinate this child’s care among the different doctors or services that this child uses?”, and “Have you felt that you could have used extra help arranging or coordinating this child’s care among the different health care providers or services?” in the past 12 months. A detailed description of the survey questions for each outcome variable is reported in the Table A1.

The independent variable of interest in the multivariable regression analysis was the variable that indicated whether the child had co-occurring ASD and epilepsy. We included intellectual disability (ID) as a main covariate due to the high rate of co-occurrence of ID, epilepsy, and ASD [19]. ID was ascertained through a question to the parent/caregiver: “Has a doctor, other health care provider EVER told you that this child has Intellectual Disability (formerly known as Mental Retardation)?” Other covariates included child age, child sex, child race (White, Black, other), insurance status (public insurance, private insurance, and uninsured), federal poverty level (FPL) (0–99% FPL, 100–199% FPL, 200–399% FPL, with 400+% FPL as reference category), and parent education level (less than high school, high school, and more than high school).

### 2.3. Statistical Analysis

Data analysis was conducted using SAS 9.4 statistical software. This analysis primarily focused on comparing the access to care indicators between the dual ASD–epilepsy group (ASD-EP) and the other two groups: children with ASD without co-occurring epilepsy (ASD-only), and children with epilepsy without co-occurring ASD (EP-only). The 10 outcome variables were first summarized and then compared between the subgroups, using a chi-square to test for significant differences. Multivariable logistic regression models were performed to determine how each covariate affects disparities in access to and utilization of health care services among children with ASD and epilepsy. Covariates used in the regression models included dual ASD-EP status, child demographics, insurance status, parent education, FPL level, and having ID. Odds ratios (OR) were used as a measure of association and 95% confidence intervals (CI) were computed. All P values were 2-sided, and a P value less than 0.05 was deemed statistically significant. Using survey procedures in SAS, all the analyses were statistically weighted to reflect the complex survey design of NSCH and produce nationally representative estimates.

## 3. Results

In total, 301,714 US children were reported to have a dual ASD-epilepsy diagnosis during 2017–2019; the unweighted sample size was 143. Table 1 shows the family and child characteristics of the study sample. Among children with ASD-EP, approximately two-thirds were aged 6–11 years old, 31% were female, 71% were White, 30% were from households with income less than 100% of the FPL, 71% had parents with above high school level of education, and 44% were publicly insured. It is notable that a substantially higher proportion of children with ASD-EP had ID (36.3%) than did children without such dual diagnosis (16.5% for ASD-only & 10.8% for EP-only).

Figure 1 compares these 10 access indicators across 3 study groups. Compared to children in the ASD-only or EP-only group, significantly more children in the ASD-EP group had a PCP (*p* = 0.0016) and a higher proportion of them needed referrals to see any doctors or receive any services (*p* = 0.0387). More parents in the ASD-EP group reported their child was admitted to the ED (*p* = 0.0001) as well as frustration with their children’s care (*p* = 0.0048); they also needed extra help with care coordination when involving different health care providers or services (*p* = 0.0039). The chi-square results did not show significant differences in several measures (e.g., USC, forgone care, inadequacy of benefits, receiving help for care coordination) among the 3 groups. Among the ASD-EP group, approximately 69% indicated that they were frustrated with their children’s medical care, 60% needed referrals to other providers, and nearly half needed extra help with coordination (Figure 1). However, nearly 88% of children in the ASD-EP group received special services.

Table 2, Table 3, Table 4, Table 5 and Table 6 presents the logistic regression analysis results. The associations between access indicators and dual ASD-EP diagnoses varied after accounting for clinical and demographic factors in the multivariable regression models. The adjusted logistic regression results in Table 2 show that children in the ASD-EP group were nearly 2.5 times more likely to visit the ED than their ASD-only counterparts (OR 2.423, 95% CI 1.254, 4.682). Health insurance type and child’s race were also associated with ED visits: privately insured children were less likely to visit the ED compared to publicly insured children (OR 0.334, 95% CI 0.202, 0.554), while Black children were about twice as likely to visit the ED compared to White children (OR 1.943, 95% CI 1.236, 3.054). Children in the ASD-EP group were 4 times more likely to have a PCP compared to children in the ASD-only group (OR 4.314, 95% CI 1.819, 10.23), while children from families of lower FPL levels (vs. FPL ≥ 400) were less likely to have a PCP (Table 2). Compared to children in the EP-only group, parents of children in the ASD-EP group were about 3 times more frustrated (OR 2.816, 95% CI 1.420, 5.582), more likely to need referral (OR 2.131, 95% CI 1.082, 4.200), and more likely to need extra help with coordinating their child’s care (OR 3.590, 95% CI 1.618, 7.967). Moreover, parents of children with ID were also more frustrated about their children’s care (OR 2.039, 95% CI 1.292, 3.219). Additionally, parents of Black children (vs. White children) or children with ID (vs. non-ID) indicated that they needed extra help for care coordination (OR 1.763, 95% CI 1.052, 2.955; OR 1.676, 95% CI 1.029, 2.731) (Table 4). Children with ID (vs. non-ID), female children (vs. male), and young children (vs. age > 5) were more likely to need referral (Table 3).

For several access indicators, demographic and clinical covariates significantly affected the access disparities among our study population regardless of having ASD-EP or not. Children from family income levels <400 FPL (vs. FPL >= 400) and children aged 12 to 17 (vs. aged 2 to 5) were more likely to encounter forgone health care (Table 5). Moreover, children from the lowest FPL level (vs. FPL >= 400) were less likely to have USC (Table 2.5) and more likely to report benefits not meeting needs (Table 5). Additionally, privately insured children (vs. publicly insured), female children (vs. male), and children from families in the lower FPL level <200 (vs. FPL >= 400) were less likely to receive developmental-related special services (Table 6). Further, parents of publicly insured (vs. privately insured) and female children (vs. male) reported that they were more likely to receive help on coordinating their children’s care among the different doctors or services (Table 4).

## 4. Discussion

Differences in access to care and healthcare utilization were found among children with ASD and epilepsy. Children with co-occurring ASD and epilepsy were more likely to have a PCP and need a referral to see any doctors or receive any services compared to children without such dual diagnoses. This means that parents had more experience with their health care providers. However, parents tended to be frustrated in their efforts to get health care services and tended to use the ED more if their child had co-occurring ASD and epilepsy; they also needed extra help with care coordination when involving different health care providers or services. When clinical characteristics and socioeconomic status were further adjusted, these differences varied depending on the specific measures of access and utilization. Having co-occurring ASD and epilepsy were strongly associated with more ED visits. Some differences in access to care and care coordination indicators were not explained by the co-occurrence of ASD and epilepsy, while they were explained by an ID diagnosis and socio-demographic factors. These factors comprised age, sex, race, insurance status, and family income levels. For example, children from poorer families were less likely to have PCP, while those who were diagnosed with ID were less likely to have benefits meeting their needs compared to those who were not diagnosed with ID.

Compared to children in the highest income level families, children in lower income level families were less likely to a have a regular place where they would usually go if sick; these parents also indicated that they were less likely to identify a personal doctor or nurse for their children. In addition, parents of children in the lowest income level were nearly three times more likely to report their insurance benefits not meeting their needs than their counterparts in the highest income level. It is notable that parents with family incomes between 200–399% FPL were more likely than parents with family incomes at or above 400 percent of FPL to forgo care for their children. Previous research also indicated that parents in middle-income families were delaying or forgoing care for their children at higher rates than those in higher-income families due to burdensome health care costs [20,21].

Among our study population, Black children were more likely to be admitted to the ED than their White counterparts after adjusting for other covariates. In addition, Black children were also less likely to receive effective care coordination. Emergency admissions indicate that Black children are experiencing ineffective epileptic control and/or inadequate coordinated care plans [12]. Our findings are consistent with the general pattern of ED disparities from other studies, which point out that the ED’s shorter wait and discharge times could be the foremost motivation for Black families’ greater usage of the ED [22,23].

Compared to male children, female children across our sample were less likely to receive special services, and they were more likely to need a referral to see any doctors or receive any services. This is consistent with a national study that found females versus males had a higher likelihood of having trouble affording needed health care [24]. Females also experience disparities in ASD-related care; female children are underserved in their access to ASD diagnoses [25]. Notably, prior work has reported that female sex is associated with higher risk of co-occurring ASD and epilepsy [9,26,27,28,29]. However, these studies did not address the access to care issue among females with ASD and epilepsy. Our findings also suggest that female children had more complex conditions that may require referral to specialist. Given the disparities in ASD-related care for females and the higher rates of co-occurring ASD and females coupled with the findings of this study, there is a clear need for more research and clinical attention on ASD and epilepsy in females.

Regardless of having a dual ASD-epilepsy diagnosis or not, parents of privately insured children were less likely to receive help on coordinating their children’s care among the different doctors or services than publicly insured children. Additionally, privately insured children were less likely to receive special services. However, having public insurance was associated with a higher likelihood of ED visits. Previous studies indicate that children with ASD with private insurance receive fewer and less adequate services than publicly insured children, and their families paid more out-of-pocket medical expenses [30,31,32].

The association between ASD, epilepsy, and ID is well recognized. Intellectual disability occurs in more than a third of children with ASD and is also the single most common risk factor for developing epilepsy [33]. It has been suggested that most of the association between epilepsy and ASD may be due to ID [34]. Our results are consistent with previous findings that ID frequently co-occurs with ASD and is also highly associated with epilepsy [27,33]. Further, our findings suggest that children with ID encountered significant barriers to health care access and coordination. Inadequate access to health services was a major problem among children with ID versus children without ID, and that contributed to the parents’ frustration about their children’s care.

For children with co-occurrence of ASD and epilepsy, there is a great need for effective care coordination with health professionals including their PCP, the ED, and referral to neurologists. The PCP’s referral decisions are crucial in the care coordination process; while a referral decision may vary according to the patients’ severity of illness level, socio-demographic factors, family preferences, and availability of specialized health professionals also play a role [15]. Coordinating care when children have ASD and epilepsy is difficult, and a lack of adequately coordinated care plans may lead to emergency admissions [12]. In addition, children with both ASD and epilepsy often require multiple providers across a continuum of specialized epilepsy care. A patient-centered, multidisciplinary approach to health care is much needed with an emphasis on improving coordination and ensuring effective communication among the PCP, specialist physicians, and families of children with co-occurring ASD and epilepsy.

There are several potential limitations to the current analysis. The NSCH is cross-sectional, so we may not infer a definitive causal association among our study variables. Both ASD and epilepsy diagnoses were based on parent report, which were not clinically validated. Additionally, while the US Census Bureau applied a nonresponse weighting adjustment to significantly mitigate any differential nonresponse, there is the potential for nonresponse bias in the NSCH as well as additional sources of bias (e.g., selection bias) that are not controlled by the weighting adjustments. Despite these limitations, this analysis identified several important inequity factors for access to care among children with ASD and epilepsy by using a large, multi-year, national database with a comprehensive set of questions on ASD and epilepsy and parents’ experiences on health care access and utilizations.

## 5. Conclusions

Our study found that having co-occurring ASD and epilepsy did not fully explain the disparities in accessing and receiving services and care coordination or experiencing parental frustration. The barriers to health care access were more likely due to the interplay of multiple clinical and individual factors. Disparities in health services use between children with and without co-occurring epilepsy were partially due to differences in children with ID and sociodemographic status. Inadequate access to health care was significantly higher among children from poor families, female children, and children with ID. Particularly, clinical decision makers and other stakeholders need greater awareness, understanding, and to act upon inequality indicators associated with low income and poverty.

## Figures and Tables

**Figure 1 children-09-01094-f001:**
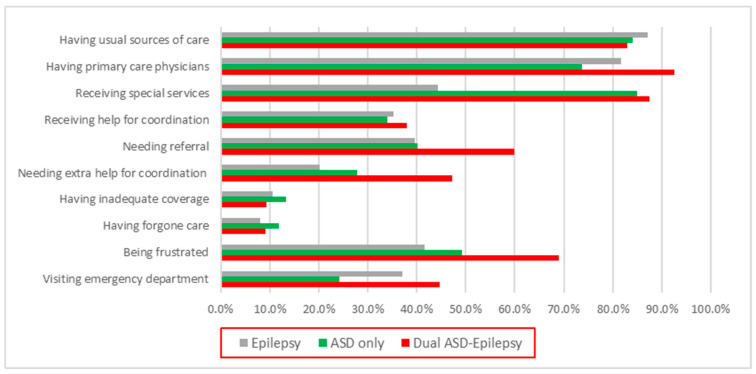
Ten access indicators among 3 groups. ASD: autism spectrum disorder.

**Table 1 children-09-01094-t001:** Family and child characteristics.

Variable	ASD N = 2150 Weighted N = 5,622,349	EP N = 711 Weighted N = 1,678,116	ASD-EP N = 143 Weighted N = 301,714
Age			
2–5	13.8%	21.7%	11.9%
6–11	69.0%	66.1%	65.8%
12–17	17.2%	12.2%	22.3%
Sex			
Male	79.8%	55.5%	69.0%
Female	20.2%	44.5%	31.0%
Race			
White	62.5%	58.8%	70.8%
Black	15.5%	20.1%	11.6%
Other	22.0%	21.0%	17.6%
Insurance Type			
Public	41.2%	34.2%	43.5%
Private	41.0%	47.9%	28.1%
Public and private	13.5%	11.7%	24.1%
Uninsured	4.3%	6.1%	4.3%
Household Poverty Level			
0–99% FPL	25.1%	28.2%	29.5%
100–199% FPL	29.4%	23.0%	24.8%
200–399% FPL	24.1%	20.6%	23.2%
400%+ FPL	21.3%	28.2%	22.5%
Parent Education Level			
Below high school	10.1%	9.5%	10.5%
High school	23.4%	18.1%	18.6%
Above high school	66.5%	72.4%	70.8%
Intellectual Disability			
Yes	16.5%	10.8%	36.3%
No	83.5%	89.2%	63.7%

Percentages are commonly rounded when presented in tables. As a result, the sum of the individual numbers does not always add up to 100%. NSCH was weighted to represent the population of children residing in US households. ASD: autism spectrum disorder; EP: epilepsy; ASD-EP: co-occurring autism spectrum disorder and epilepsy; FPL: federal poverty level.

**Table 2 children-09-01094-t002:** Adjusted odds ratios for visiting ED and having PCP.

Variable.	Visiting Emergency Department (ED)		Having Primary Care Physicians (PCP)	
	**Odds Ratio (95% CI)**	** *p* **	**Odds Ratio (95% CI)**	** *p* **
**ASD-EP**				
ASD-EP (vs. EP only)	1.290 (0.651–2.557)	0.4661	2.511 (1.018–6.196)	0.0457
ASD-EP (vs. ASD only)	2.423 (1.254–4.682)	0.0084	4.314 (1.819–10.23)	0.0009
**Age**				
6–11 (vs. 2–5)	0.606 (0.388–0.946)	0.0276	1.039 (0.625–1.726)	0.8839
12–17 (vs. 2–5)	0.670 (0.382–1.175)	0.1622	1.120 (0.544–2.308)	0.7576
**Sex**				
Female (vs. male)	1.031 (0.690–1.542)	0.8798	1.277 (0.760–2.146)	0.3564
**Race**				
Black (vs. White)	1.943 (1.236–3.054)	0.0040	0.701 (0.398–1.237)	0.2201
Other (vs. White)	0.945 (0.514–1.736)	0.8550	0.623 (0.340–1.141)	0.1253
**Insurance Type**				
Private (vs. public)	0.334 (0.202–0.554)	<0.0001	0.662 (0.386–1.134)	0.1328
**Household Poverty Level**				
0–99% FPL (vs. 400+)	1.034 (0.537–1.991)	0.9192	0.305 (0.145–0.645)	0.0019
100–199% FPL (vs. 400+)	0.797 (0.494–1.285)	0.3520	0.367 (0.203–0.665)	0.0009
200–399% FPL (vs. 400+)	1.316 (0.893–1.940)	0.1647	0.485 (0.297–0.794)	0.0040
**Parent Education Level**				
High school (vs. below)	0.611 (0.247–1.512)	0.2861	2.033 (0.691–5.982)	0.1974
Above high school (vs. below)	0.900 (0.342–2.366)	0.8309	2.080 (0.728–5.940)	0.1714
**Intellectual Disability (ID)**				
Yes (vs. no)	1.095 (0.681–1.762)	0.7075	0.916 (0.476–1.764)	0.7932

Source: 2017–2019 National Survey for Children’s Health. ED: emergency department; PCP: primary care physicians.

**Table 3 children-09-01094-t003:** Adjusted odds ratios for parental frustration and needing referrals.

Variable	Parental Frustration		Needing Referral	
	**Odds Ratio (95% CI)**	** *p* **	**Odds Ratio (95% CI)**	** *p* **
**ASD-EP**				
ASD–EP (vs. EP only)	2.816 (1.420–5.582)	0.0030	2.131 (1.082–4.200)	0.0287
ASD–EP (vs. ASD only)	2.035 (1.072–3.863)	0.0298	1.735 (0.913–3.299)	0.0926
**Age**				
6–11 (vs. 2–5)	0.960 (0.650–1.432)	0.8590	0.470 (0.313–0.705)	0.0003
12–17 (vs. 2–5)	0.812 (0.458–1.442)	0.4773	0.518 (0.295–0.907)	0.0215
**Sex**				
Female (vs. male)	1.196 (0.835–1.715)	0.3292	1.661 (1.187–2.326)	0.0031
**Race**				
Black (vs. White)	0.917 (0.589–1.430)	0.7035	1.466 (0.925–2.323)	0.1036
Other (vs. White)	0.678 (0.401–1.147)	0.1477	0.712 (0.448–1.133)	0.1515
**Insurance Type**				
Private (vs. public)	0.864 (0.523–1.426)	0.5666	0.727 (0.473–1.117)	0.1456
**Household Poverty Level**				
0–99% FPL (vs. 400+)	0.802 (0.467–1.378)	0.4243	1.327 (0.757–2.325)	0.3235
100–199% FPL (vs. 400+)	0.706 (0.427–1.168)	0.1751	1.077 (0.675–1.720)	0.7552
200–399% FPL (vs. 400+)	1.184 (0.846–1.657)	0.3251	1.573 (1.114–2.221)	0.0100
**Parent Education Level**				
High school (vs. below)	1.067 (0.428–2.662)	0.8889	1.428 (0.564–3.616)	0.4519
Above high school (vs. below)	0.781 (0.331–1.844)	0.5730	1.833 (0.745–4.507)	0.1870
**Intellectual Disability (ID)**				
Yes (vs. no)	2.039 (1.292–3.219)	0.0022	1.872 (1.190–2.944)	0.0067

Source: 2017–2019 National Survey for Children’s Health.

**Table 4 children-09-01094-t004:** Adjusted odds ratios for care coordination indicators.

Variable	Needing Extra Help		Receiving Help	
	**Odds Ratio (95% CI)**	** *p* **	**Odds Ratio (95% CI)**	** *p* **
**ASD-EP**				
ASD–EP (vs. EP only)	3.590 (1.618–7.967)	0.0017	1.034 (0.511–2.091)	0.9257
ASD–EP (vs. ASD only)	1.998 (1.054–3.787)	0.0338	1.020 (0.525–1.983)	0.9528
**Age**				
6–11 (vs. 2–5)	0.834 (0.535–1.301)	0.4234	0.449 (0.297–0.679)	0.0002
12–17 (vs. 2–5)	0.798 (0.446–1.425)	0.4452	0.733 (0.421–1.275)	0.2706
**Sex**				
Female (vs. male)	1.556 (0.974–2.485)	0.0641	1.754 (1.190–2.585)	0.0045
**Race**				
Black (vs. White)	1.763 (1.052–2.955)	0.0314	0.906 (0.529–1.549)	0.7170
Other (vs. White)	0.798 (0.446–1.425)	0.9490	0.856 (0.530–1.385)	0.5271
**Insurance Type**				
Private (vs. public)	0.847 (0.505–1.419)	0.5274	0.448 (0.266–0.755)	0.0026
**Household Poverty Level**				
0–99% FPL (vs. 400+)	0.996 (0.518–1.917)	0.9909	0.782 (0.426–1.434)	0.4259
100–199% FPL (vs. 400+)	0.706 (0.410–1.217)	0.2106	1.121 (0.672–1.871)	0.6614
200–399% FPL (vs. 400+)	1.525 (0.992–2.344)	0.0547	1.081 (0.722–1.616)	0.7060
**Parent Education Level**				
High school (vs. below)	0.877 (0.262–2.936)	0.8312	0.973 (0.325–2.915)	0.9613
Above high school (vs. below)	1.119 (0.335–3.732)	0.8549	0.787 (0.274–2.263)	0.6570
**Intellectual Disability (ID)**				
Yes (vs. no)	1.676 (1.029–2.731)	0.0381	1.156 (0.697–1.917)	0.5746

Source: 2017–2019 National Survey for Children’s Health.

**Table 5 children-09-01094-t005:** Adjusted odds ratios for forgone care and inadequacy of benefits.

Variable	Forgone Health Care		Inadequacy of Benefits	
	**Odds Ratio (95% CI)**	** *p* **	**Odds Ratio (95% CI)**	** *p* **
**ASD-EP**				
ASD–EP (vs. EP only)	0.928 (0.367–2.348)	0.8749	1.229 (0.484–3.118)	0.6644
ASD–EP (vs. ASD only)	0.653 (0.283–1.506)	0.3172	1.649 (0.719–3.781)	0.2378
**Age**				
6–11 (vs. 2–5)	1.636 (0.838–3.193)	0.1493	0.802 (0.457–1.405)	0.4400
12–17 (vs. 2–5)	2.694 (1.087–6.677)	0.0324	0.712 (0.336–1.507)	0.3741
**Sex**				
Female (vs. male)	1.114 (0.622–1.996)	0.7163	0.601 (0.341–1.057)	0.0772
**Race**				
Black (vs. White)	0.835 (0.382–1.826)	0.6517	1.570 (0.745–3.309)	0.2360
Other (vs. White)	1.585 (0.732–3.431)	0.2420	0.675 (0.377–1.209)	0.1859
**Insurance Type**				
Private (vs. public)	1.443 (0.621–3.352)	0.3940	0.815 (0.454–1.464)	0.4934
**Household Poverty Level**				
0–99% FPL (vs. 400+)	2.252 (1.055–4.806)	0.0358	2.771 (1.075–7.140)	0.0349
100–199% FPL (vs. 400+)	3.232 (1.432–7.294)	0.0047	0.919 (0.487–1.736)	0.7953
200–399% FPL (vs. 400+)	2.662 (1.418–4.999)	0.0023	0.928 (0.562–1.532)	0.7693
**Parent Education Level**				
High school (vs. below)	1.379 (0.407–4.671)	0.6057	4.854 (1.386–16.99)	0.0135
Above high school (vs. below)	1.458 (0.482–4.411)	0.5043	3.235 (1.027–10.19)	0.0449
**Intellectual Disability (ID)**				
Yes (vs. no)	1.793 (0.990–3.249)	0.0541	0.537 (0.294–0.980)	0.0427

Source: 2017–2019 National Survey for Children’s Health.

**Table 6 children-09-01094-t006:** Adjusted odds ratios for USC and special services.

Variable	Having Usual Sources of Care (USC)		Receiving Special Services	
	**Odds Ratio (95% CI)**	** *p* **	**Odds Ratio (95% CI)**	** *p* **
**ASD-EP**				
ASD–EP (vs. EP only)	0.777 (0.314–1.923)	0.5846	8.201 (2.712–24.80)	0.0002
ASD–EP (vs. ASD only)	1.057 (0.453–2.467)	0.8979	1.112 (0.365–3.383)	0.8519
**Age**				
6–11 (vs. 2–5)	0.942 (0.508–1.748)	0.8504	0.643 (0.422–0.980)	0.0400
12–17 (vs. 2–5)	0.631 (0.286–1.391)	0.2534	0.319 (0.165–0.617)	0.0007
**Sex**				
Female (vs. male)	0.917 (0.571–1.471)	0.7178	0.575 (0.382–0.866)	0.0081
**Race**				
Black (vs. White)	0.840 (0.446–1.581)	0.5890	1.791 (0.999–3.209)	0.0503
Other (vs. White)	0.874 (0.480–1.593)	0.6610	1.612 (0.893–2.908)	0.1128
**Insurance Type**				
Private (vs. public)	0.959 (0.510–1.802)	0.8964	0.601 (0.368–0.981)	0.0416
**Household Poverty Level**				
0–99% FPL (vs. 400+)	0.282 (0.129–0.620)	0.0016	0.430 (0.217–0.854)	0.0159
100–199% FPL (vs. 400+)	0.366 (0.186–0.722)	0.0037	0.521 (0.302–0.900)	0.0193
200–399% FPL (vs. 400+)	0.482 (0.271–0.855)	0.0127	0.819 (0.539–1.243)	0.3474
**Parent Education Level**				
High school (vs. below)	0.546 (0.182–1.641)	0.2812	0.785 (0.291–2.118)	0.6323
Above high school (vs. below)	0.522 (0.178–1.531)	0.2363	0.594 (0.233–1.514)	0.2749
**Intellectual Disability (ID)**				
Yes (vs. no)	1.110 (0.599–2.057)	0.7396	4.312 (1.653–11.25)	0.0028

Source: 2017–2019 National Survey for Children’s Health.

## Data Availability

The data that support the findings of this study are available from the corresponding author upon reasonable request.

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
