# Peer review of "Access to Care for US Children with Co-Occurrence of Autism Spectrum Disorder and Epilepsy"

_children, 2022, doi:10.3390/children9071094_

Round 1

Reviewer 1 Report

The research problem presented by the authors of the article relates to an important issue of access to health care for children with autism spectrum disorder (ASD) and epilepsy (EP). Access to healthcare appears in this study as a dependent variable, and the independent variables included factors such as ASD-EP status, child demographics and intellectual disability. The analysis of the research results showed that, although having co-occurring ASD and epilepsy did not fully explain the disparities in accessing and receiving services and care coordination, however, inadequate access to healthcare was significantly higher among female children, children from poor families, and children with intellectual disability. The authors are aware of certain limitations related to the analysis of the obtained results, which positively proves their research reliability. The most important effect of these studies is the identification of several important inequity factors for access to care among children with ASD and epilepsy, which are present even in economically highly developed countries such as the United States. Taking into account the significant scientific and informational value of the obtained results, I support publishing the reviewed article.

Author Response

We appreciate the reviewer's supportive comments!

Reviewer 2 Report

Dear Authors,

Thank you for the opportunity to review this paper.
The message of the paper is clear, the article has a logical structure, and it is interesting to read.
I have the following comments and suggestions:

Lines 35-36
Your statement .... epilepsy occurs in 12.5% of children under age 17 – this statement is not fully
accurate. Reference (10) says that ..... The average prevalence of epilepsy in children with ASD 2-17
years was 12.5%. Since in the previous sentence you refer to a review indicating a review 2.4% to
26% prevalence of epilepsy in ASD, I suggest to mention that the 12,5% prevalence was observed in a
population-based survey.

Lines 96-98
A suggestion: Since non-US readers will be reading this paper, it would be helpful to briefly explain
the meaning of terms. E.g. what is the principal difference between having usual sources of care
(USC) and having primary care physicians (PCP). Later in the text, you speak about the referral to a
specialist. A referral to a specialist is routinely requested in many countries; therefore it is not fully
clear why it is presented/perceived in the paper as a negative (at least this is my perception). This
explanation may be useful for a better understanding the context. An option is to briefly discuss this
issue in the Discussion section.

Line 100
Please state what “otherwise“ means. Is it the answer “no”, or missing answer, or another option?

Line 113
Please state how the ID was detected. Was it a similar question as asking for ASD and for epilepsy?
Later in text you refer that is an explanatory variable

Lines 136-137
Please rephrase the sentence. Your primary result in your sample is n=143. Which implies that at the
national level 301,714 children suffer from this condition.

Table 1
Although described in the methods, it would be helpful to the reader if the meaning of “weighted N”
is explained e.g. as a comment below the table.
Please replace the figure with a graphic with a better resolution.

Line 145 – 156
There is no comment to the variable “Having foregone care” in the text, although it is presented in
the Figure 1. Please indicate in the text if the additional variables have/ or have not displayed
significant differences.

Lines 156 – Fig 1
For the reader it is not fully clear why you have decided to include “Having Foregone Care” into Fig 1
(no significant difference in Odds ratio in the 3 groups, tab 2. 4), and you have not included “Special
services” (significant difference in Odds ratios in ASD-EP vs EP, tab 2.5). All variables presented in Fig
1 except “Having Foregone Care” display significant differences at least between 2 groups,
sometimes even between all groups. Moreover, “Having foregone care” applies to a small percent of
participants. Could you please explain or reconsider the choice of variables.

Line 202
You state that “Children with co-occurring ASD and epilepsy were more likely to have a personal
doctor or nurse”. Please specify which of the variables indicates this. Does this fact mean extra
payments for health services? The readers who are not much familiar with the US health care system
may not be clear if this is a benefit or a negative.

General comments
- Please avoid using abbreviations in tables and figures. If unavoidable, explain the
abbreviations below the table.
- In fact all characteristics except Parent Educational Level have significant associations with
indicators of the access to healthcare. This is clearly discussed in the Discussion section.
However, it would be good to summarize, if according to your expert opinion any of them is
more significant than others.
- It is upon decision of the Editorial Board if any type of Ethical statement is needed in this
type of paper.

Author Response

We appreciate the reviewer's supportive comments. Please see authors' responses in the attached pdf document.
